**Data Availability Statement:** All relevant data are within the paper and its Supporting Information files.

**Funding:** The authors received no specific funding for this work.

# In-silico modelling of the impact of hypertension on the mean transvalvular gradients in aortic stenosis

**Jacques Liebenberg**[1]*, **Anton Doubell**[1], **Ryno Laubscher**[2], **Philip Herbst**[1]

1 Division of Cardiology, Department of Medicine, Tygerberg Hospital and University of Stellenbosch, Stellenbosch, South Africa, 2 Computational Fluid Dynamics and Scientific Machine Learning, Department of Mechanical and Mechatronic Engineering, University of Stellenbosch, Stellenbosch, South Africa

* liebjurg@gmail.com

## Abstract

### Introduction

The influence of hypertension on the diagnostic assessment of aortic stenosis (AS) severity is unclear, yet clinically relevant. To clarify the effect of hypertension on transvalvular gradients, requires a better understanding of the impact that blood pressure change has on mean flow rate. Also, the effect of various degrees of AS severity, the valve geometry and intrinsic left ventricular contractile function (elastance) on this interaction, needs to be clarified. The current work aims to assess this interaction and the magnitude of these effects.

### Methods

A validated, zero-dimensional electro-hydraulic analogue computer model of the human cardiovascular circulatory system was generated. It was used to assess the impact of blood pressure changes on left ventricular pressure and transvalvular gradients at various flow rates, left ventricular elastances, a range of aortic valve areas and for different aortic valve morphologies.

### Results and discussion

The magnitude of the impact of hypertension induced changes on the mean gradient (MG) is influenced by the mean flow rate, the AS severity, the hydraulic effective valve orifice area and the left ventricular elastance. Generally, for a given change in systemic arterial pressure, the impact on MG will be the most marked for lower flow rate states such as is expected in more severe degrees of AS, for worse intrinsic left ventricular (LV) contractility, shorter ejection times and lower end diastolic LV volumes. Given the above conditions, the magnitude of the effect will be more for a larger aortic sinus diameter, and also for a typical degenerative valve morphology compared to a typical rheumatic valve morphology.

**Competing interests:** The authors have declared that no competing interests exist.

## Conclusion

The interaction between hypertension and mean gradients in AS is complex. The current work places previous recommendations in perspective by quantifying the magnitude of the effect that the changes in blood pressure has on mean gradient in various pathophysiological states. The work creates a framework for the parameters that should be considered in future clinical research on the topic.

## Introduction

Mean global life expectancy is increasing as a result of urbanization, wider access to medical care and advances in medical technology [1]. This has resulted in a worldwide shift towards conditions associated with advancing age, including degenerative aortic stenosis (AS). Over the past two decades and with the advent of trans aortic valve implantation (TAVI), many patient cohorts previously deemed unfit for surgery, are now offered valve intervention. This expansion of treatable patients has renewed the focus on the diagnostic assessment of AS and has highlighted some discrepancies in the parameters we use to classify AS severity, specifically in patients with low transvalvular gradients and in patients with an impaired ejection fraction. This, often difficult diagnostic evaluation, is frequently complicated by the presence of concomitant hypertension which affects 78% of patients suffering from AS [2]. The physics underlying and the response behavior of hypertension induced changes in mean transvalvular gradients in AS across the spectrum of ejection fraction and transvalvular gradients has not previously been fully investigated.

Transvalvular gradients (or the pressure drop across the aortic valve), which forms the cornerstone of the echocardiographic diagnostic assessment of AS severity, is often thought of as the numerical difference between the left ventricular pressure and the blood pressure (BP). This understanding has its root in the physics principle that $\Delta P = P2-P1$ (P = pressure). This construct, which has merits, is at first glance supported by the Zva (valvuloarterial impedance) concept, which views the global LV afterload as the sum of sequential constrictors in series and depicts LVSP = MG plus SAP (LVSP = Left ventricular systolic pressure, MG = mean transvalvular gradient, and SAP = systolic arterial pressure). This reductionist viewpoint of the relationship between LV pressure, BP and MG may lead one to believe that for any degree of rise/fall in SAP, a numerically similar but directionally opposite impact on mean transvalvular gradients should result. Even leading textbooks in cardiology states that a reduction in peripheral vascular resistance will result in an increase in transvalvular gradients, and that this may precipitate symptoms [3]. Such statements provide evidence for the extent of this oversimplified view.

However, from a flow dynamics theory point of view, Newton's second law gives us insight into the factors that govern transvalvular gradients. This relationship can be depicted as $\Delta P = \kappa \frac{Q^2}{AVA^2}$ where $\Delta P$ = transvalvular gradient, $\kappa$ = combined energy loss coefficient, Q = mean flow (ml/s), and AVA = aortic valve area (cm$^2$). It is important to note that, the absolute pressure difference between the LV and the arterial system does not feature in this equation. Furthermore, for a given AVA, MG is therefore entirely dependent on the transvalvular flow and the combined energy loss coefficient.

To understand what the impact of hypertension is on transvalvular gradients, it is therefore necessary to understand what the impact of hypertension is on the mean transvalvular flow

rate (MFR). The flow dependance of hypertension induced changes on MFR and hence on transvalvular gradients have previously been documented [4–8].

Mascherbauer et al. [4] demonstrated that the transvalvular gradients for a valve area of 1.0 cm$^2$ remained unchanged across a wide spectrum of BP alterations in an in-vitro fixed plate model. The authors did however rightfully state that their in-vitro testing may not have accounted for in-vivo changes in MFR that may take place with alterations in BP.

Kadem et al. [5] induced AS by supravalvular banding of porcine aortas, followed by pharmacological induction of hypertension. Their work suggested that concomitant hypertension may lead to diminished transvalvular flow and gradients and hence lead to underappreciation of AS severity. The suggested mechanisms were a) a decrease in the MFR as a result of the increased afterload even though a change in MFR was not demonstrated in their work, making this a very improbable mechanism, b) an increase in the effective orifice area due to splinting/pressurization of the sinuses below the banded aorta, an effect that is likely only relevant in supra-aortic banding with intrinsically normal and pliable leaflets. It should also be noted that supravalvular banding of the aorta represents combined loss coefficients that are vastly different from that of a calcified aortic valve. Furthermore, the strategy of acutely inducing AS, as was done by supravalvular banding, is a crude simulation of longstanding and progressive calcific AS. This casts doubt on the clinical applicability of these findings in humans with aortic calcification and severe calcific AS.

Côté et al. [6] made important contributions to the field in their animal studies and observational in–vivo studies. They reported a reduction in MG in hypertensive states and with reduced arterial compliance, despite a constant MFR. The suggested mechanism is thought to be related to reduced arterial compliance in hypertension. A recent study by Laubscher et al. [7] evaluated the relative contributions of the components of combined loss coefficients. Laubscher et al. reported that sudden expansion losses are the primary determinant of the pressure loss, and that arterial elastance plays a negligible role, largely in contradiction to the findings of Côté et al. These conflicting findings highlight the strengths and limitations of in–vivo and mathematical modeling studies. It should be noted that, in the Côté et al. work, the 95% confidence intervals for the MG for each of the 4 specified comparative subgroups overlapped, a commonly encountered limitation of the non-idealized flow of in–vivo studies resulting in wide uncertainty intervals.

An article by Orłowska-Baranowska [8] describes the flow dependance of gradients but highlights discrepant findings in previous publications as to the absolute magnitude of the impact of hypertension induced changes on mean transvalvular flow, and hence transvalvular gradients.

A hiatus in the existing literature is the lack of consideration for the full complement of factors that influences and governs the intricate relationship between MG and SAP. For example, no previous in–vivo or clinical study have reported on the intrinsic contractile state of the LV, an important determinant of mean transvalvular flow, and how this influences BP mediated changes in flow rate and transvalvular gradients. Certainly, for a given change in BP, the subsequent change in MFR and the relationship that such change in MFR has with the MG, will be vastly different in mild vs critical AS; in a normotensive state vs in severe hypertension; but also importantly, in myopathic ventricles compared to ventricles with preserved intrinsic contractile strength.

Whereas the directionality of change in MG with changes in SAP is not debated, it is the quantification and characterization of the MFR and MG response behavior over a wide spectrum of global LV afterload's (including the absolute AS severity and degree of hypertension) as well as LV contractile states that is the focus of the current work (Fig 1).

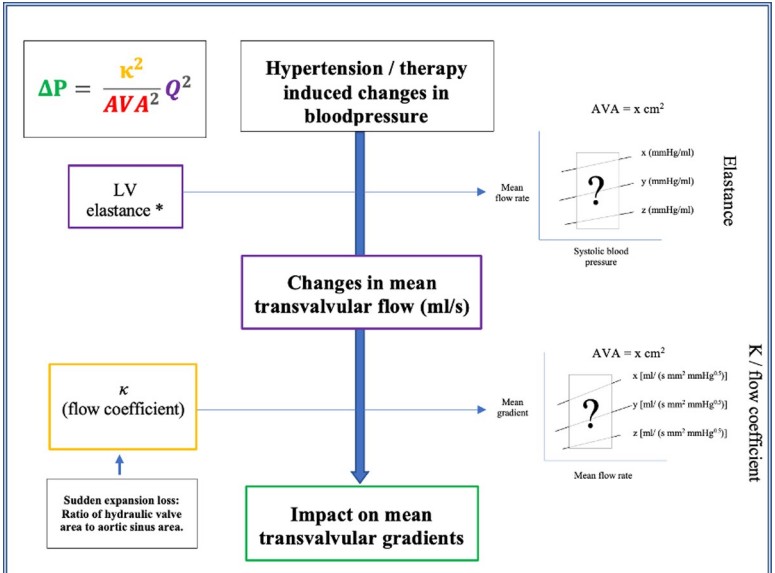

**Fig 1. Central illustration demonstrating the complex relationship between systolic blood pressure and mean transvalvular gradients.** For a given AVA, the impact of hypertension on MG is mediated through a change in mean flow rate (MFR). The magnitude of this impact is modulated by the LV elastance. Subsequently, the magnitude of MFR induced changes in MG is modulated by the combined energy loss coefficient. AVA–aortic valve area, LV–left ventricle. * Other factors that influences the MFR and the vulnerability to blood pressure induced changes in MFR include preload (left ventricular end diastolic volume); and the end-systolic activation time of the ventricles, systemic arterial compliance, and right ventricular unstressed volume.

## Methods

To investigate the impact of changes in systolic BP on the mean transvalvular gradient in varying degrees of AS and at varying flow states, a zero-dimensional electro-hydraulic analogue computer model of the human cardiovascular circulatory system was used (see Fig 2 and S1 File). Pressure and flow rates were simulated throughout the system by using a computer-based mathematical model that solves a set of ordinary differential equations (ODEs) and various constitutive relations [9]. The current modeling is built on previously validated data that have been shown to replicate actual cardiovascular behavior well and have been validated using actual patient data [10]. A schematic representation of the model can be seen in Fig 3.

In this model, the contractility and pressure-volume relationships of the cardiac chambers are approximated using the time-varying elastance model previously described by Suga et al. [11] and the various compartments that make up the systemic and pulmonary vasculature was modelled using resistance, inertial and capacitance components as seen in Fig 2. Heart valves were modelled as area-varying orifice components. For a detailed review of references of validation studies, and a review of the model development, the input variables, baseline assumptions, and the set boundary conditions, please refer to S1 File.

Using the above model, MFR response behavior was characterized over a spectrum of systolic BP's for a range of AS (ranging from very severe to mild) and over a range of LV elastances. AS severities were defined as follow: very severe—$0.8cm^2$; severe -$1.0cm^2$, moderate—$1.5cm^2$; and normal– 4. $5cm^2$. The range of LV elastances tested included LV elastances 2.5, 1.5 and 1.25 mmHg/ml simulating a range of intrinsic LV contractile strength from normal to a markedly impaired. Following this, the impact of a change in MFR on the MG was assessed for each of the different degrees of AS. From this we determined the MFR behavior over a spectrum of afterloads. After having characterized the SAP–MFR response behavior for each of the

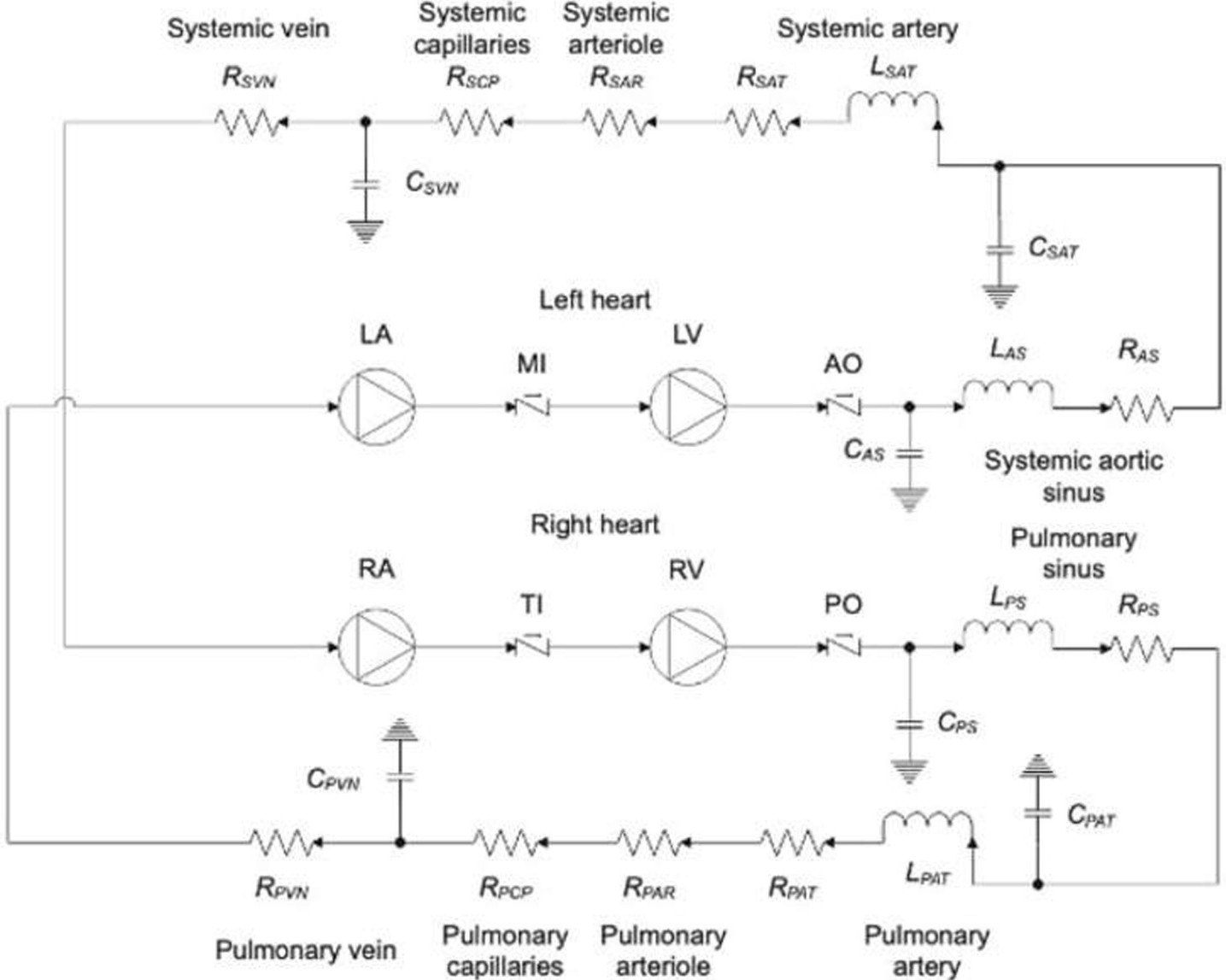

**Fig 2. Zero—dimensional electro-hydraulic analogue model of human cardiovascular system.** (R–resistance; L–inductance; C–capacitance; LA–left atrium; MI–mitral valve; LV–left ventricle; AO–aortic valve; RA–right atrium; TI–tricuspid valve; RV–right ventricle; PO–pulmonary valve).

AVA's and at varies LV elastances, we then assessed the MFR–MG response behavior. By doing this, we were able to simulate various clinical scenarios and to predict and/or quantify the absolute changes in MG per unit change in SAP.

## Results

The results in Table 1 illustrate that the lower the LV elastance, the more vulnerable the LV is to a decrease in MFR as a result of an increase in SAP. For an AVA of 4.5cm$^2$, the ratio of ΔSAP: ΔMFR was 93mmHg: 10ml/s for the elastance of 2.5mmHg/ml, whereas the ratio was 58mmHg: 25ml/s for the elastance of 1.25mmHg/ml. The same pattern was observed for the 1.5cm$^2$, 1.0cm$^2$, and the 0.8cm$^2$ valve areas.

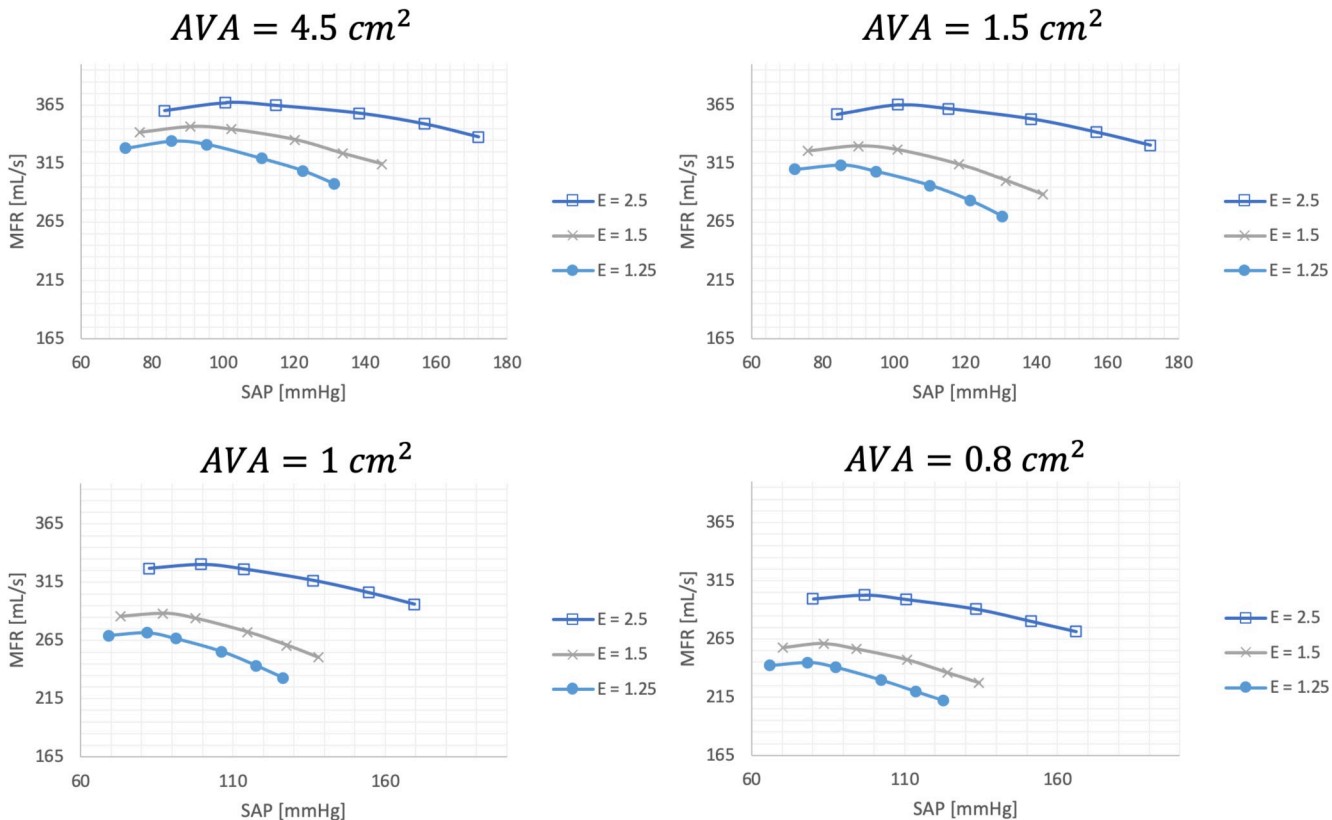

**Fig 3. Mean flow rate (ml/s) response behavior over a range of systolic arterial blood pressures (mmHg) for various aortic valve area's (AVA) and various LV elastances (E).**

**Table 1. Baseline experiment findings.**

| AV–area (cm²) | LV Elastance (mmHg/ml) | Baseline | | | | Post simulated intervention | | | |
|---|---|---|---|---|---|---|---|---|---|
| | | SAP (mmHg) | LVSP (mmHg) | MG (mmHg) | MFR (ml//s) | SAP (mmHg) | LVSP (mmHg) | MG (mmHg) | MFR (ml//s) |
| 4.5cm² | 2.5 | 91 | 92 | 1.5 | 299 | 184 | 184 | 1.4 | 289 |
| | 1.5 | 82 | 82 | 1.1 | 276 | 150 | 150 | 1.0 | 256 |
| | 1.25 | 79 | 79 | 1.0 | 265 | 137 | 137 | 0.9 | 240 |
| 1.5cm² | 2.5 | 93 | 109 | 16.8 | 298 | 184 | 194 | 15.0 | 287 |
| | 1.5 | 83 | 92 | 13.0 | 272 | 150 | 155 | 11.0 | 252 |
| | 1.25 | 79 | 86 | 11.6 | 260 | 137 | 141 | 9.4 | 236 |
| 1.0cm² | 2.5 | 91 | 143 | 43 | 279 | 181 | 219 | 37 | 260 |
| | 1.5 | 80 | 114 | 32.2 | 241 | 145 | 169 | 25.9 | 219 |
| | 1.25 | 75 | 105 | 29 | 228 | 152 | 132 | 22 | 192 |
| 0.8cm² | 2.5 | 86 | 173 | 67.4 | 245 | 172 | 238 | 57.1 | 228 |
| | 1.5 | 73 | 133 | 48 | 207 | 136 | 179 | 38.8 | 188 |
| | 1.25 | 69 | 121 | 42.4 | 195 | 124 | 160 | 33.0 | 173 |

AV–aortic valve; LV–left ventricle; SAP–systolic arterial pressure; LVSP–left ventricular systolic pressure; MG–mean gradient; MFR–mean transvalvular flow rate

Similarly, the more severe the AS (smaller AVA), the more vulnerable the LV is to a decrease in MFR as a result of an increase in SAP. For an elastance of 2.5mmHg/ml, the ratio of ΔSAP: ΔMFR was 93mmHg:10ml/s for the AVA of 4.5cm$^2$, whereas the ratio was 86mmHg: 17ml/s for the AVA of 0.8cm$^2$. The same pattern was observed for the elastances of 1.5mmHg/ml and 1.25mmHg/ml.

The general trend therefore is clear: With progressively more severe AS, and with a lower LV elastance, the MFR will fall more for any given increase in SAP.

Fig 3 illustrates the MFR response behavior as a function of SAP for various AVA's and for various LV elastances. It is interesting to note that for non-severe AS (AVA 4.5cm$^2$ & 1.5cm$^2$), the flow rate response behavior is largely similar, and for the severe and very severe AS, the flow rate responses appear similar. Furthermore, the general trend is that the MFR remains relatively stable over an initial SAP range, whereafter an inflection point occurs and that for subsequent increases in SAP, the MFR decreases. The absolute SAP at which this inflection point occurs, as well as the steepness of the subsequent decline in MFR, depends on the AVA as well as the elastance. For illustrative purposes, with a preserved LV elastance (2.5mmHg/ml), and an AVA of 1.5cm$^2$, an increase in SAP from 120mmHg to 160mmHg (+40mmHg) results in a decrease in MFR of 20ml/s. Conversely, at a smaller AVA of 0.8cm$^2$ and a worse elastance of 1.25mmHg/ml a smaller increase in SAP from 80mHg to 110mmHg (+30mmHg) is required to result in a similar fall in MFR of 20ml/s.

Next, we characterized the MG–MFR response behavior for each of the AVAs using a constant $\kappa$—value of 240 ml/(s mm$^2$Hg$^{0.5}$).

From Fig 4, it is demonstrated that the LV elastance appears to have no independent impact on the MG (elastance slopes virtually superimposable for each of the AVA's), but that it rather manifests its impact on the MG through its impact on MFR. The superimposable MG—MFR response curves for the various elastances at a given AVA and at a constant energy loss coefficient provides support for the interaction between hypertension and mean transvalvular gradients as depicted in the central illustration (Fig 1).

Next, we explored the impact of a change in $\kappa$ on the MG—MFR relationship. In an earlier publication by Laubscher et al. it was shown that the majority of the variation in the energy loss coefficient is due to sudden expansion losses as blood exits the leaflet orifice and enters the aortic sinuses. The magnitude of the energy loss coefficient due to sudden expansion losses is driven by the ratio of the AVA to the area of the aortic sinus. For explorative purposes, we modelled the MG—MFR relationship for the AVA of 0.8cm$^2$ at a $\kappa$-value as determined by a sinus dimension of 25mm, 30mm, and 35mm respectively. (See Fig 5).

A final layer of complexity to consider in assessing and understanding the impact of hypertension on the MG in AS, is the difference between the hemodynamic valve area and the hydraulic valve area. Echocardiographic data typically reveals a hemodynamic valve area, but this does not consider the morphology of the orifice. The hydraulic area is calculated as the ratio of the cross-sectional area of the flow channel to the length of the wetted perimeter. The valve morphology determines the perimeter of the orifice area and in doing so influences the hydraulic area. All calculations in this manuscript have thus far considered sudden expansion losses based on a circular orifice area (approximating the expected morphology of a rheumatic aortic valve). For comparison purposes we have modelled the MG—MFR relationship for a valve area of 0.8cm$^2$, with a constant sinus diameter of 30mm, but with a $\kappa$ value as determined by a valve perimeter of 3.17cm vs 3.53cm. In addition to have a larger range of flow rates the systemic arterial compliance, end-systolic ventricular activation time and right ventricle unstressed volume was changed to allow for higher MFR. See Fig 6A and 6B.

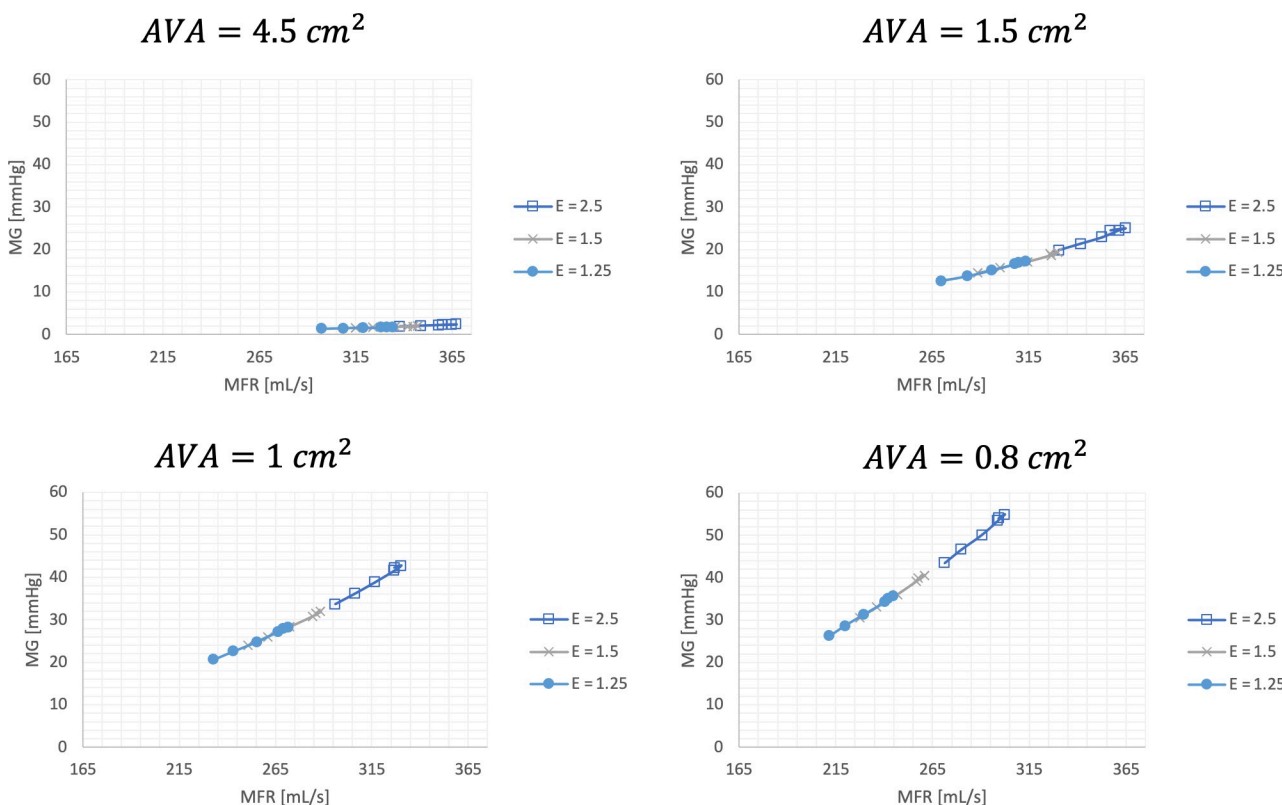

**Fig 4. Mean aortic transvalvular gradient (mmHg) (Y-axis) over a range of mean flow rate (ml/s) (X–axis).**

## Discussion

The results of the current work places the known impact of hypertension on the mean transvalvular gradients in AS in perspective by comprehensive in-silico modelling that incooperates all the hemodynamic factors that influence this intricate relationship. We have demonstrated that the predominant impact of hypertension on MG is mediated through changes in MFR, and that magnitude of the effect is modulated by the degree of AS as well as by the intrinsic LV contractile strength, the left ventricular diastolic volume, and the ejection time. Furthermore, the combined energy loss coefficient that is primarily determined by the valve orifice morphology and ratio of the hydraulic orifice area to sinus diameter, also have an impact on the magnitude of change in MG for a given change in MFR. Generally, for a given change in SAP, the impact on MFR and hence MG will be the most marked for lower flow rate states such as is expected in more severe degrees of AS, for worse intrinsic LV contractility, shorter ejection times and lower end diastolic LV volumes. For the said conditions, the magnitude of the effect will likely be more the larger the aortic sinus diameter, and also with a typical degenerative morphology compared to a typical rheumatic valve morphology. Various combinations of these conditions can result in a range of changes in MG when BP is altered, from minor/none to very large and clinically significant changes.

The novel contribution of the current work, in contrast to previous publications, is that it uses validated flow dynamics modelling grounded in fundamental physics principles to develop a digital platform where various parameters can each be manipulated to quantify effect size. Whereas mathematical modelling studies cannot be used to inform clinical decision making, they do form the basic sciences groundwork for clinical studies. The existing clinical

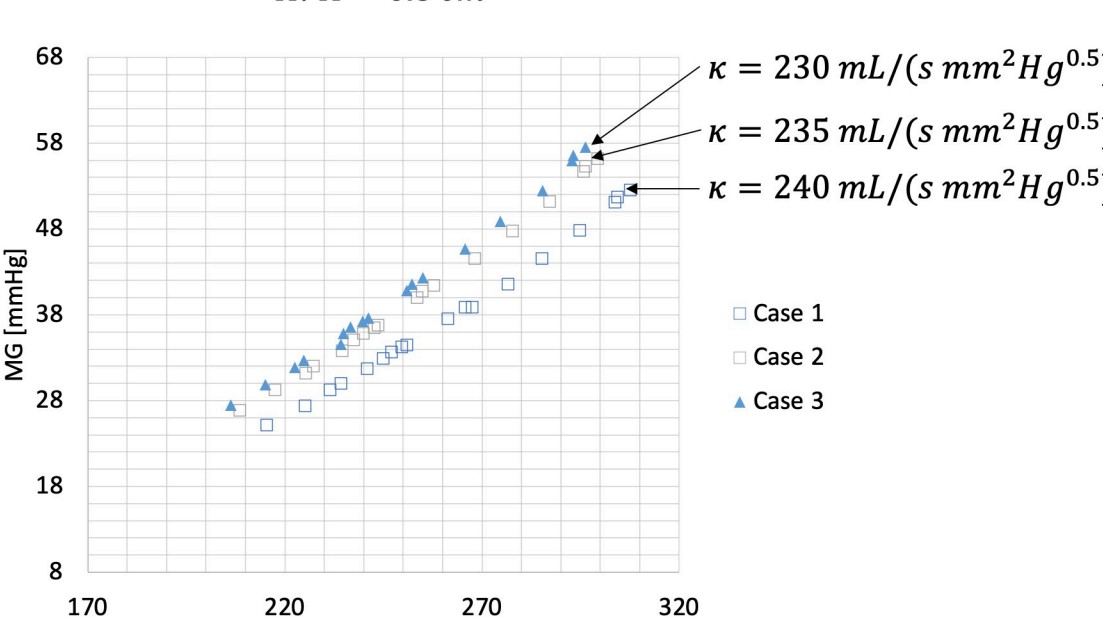

$$AVA = 0.8\ cm^2$$

Case 1: $D_{sinus} = 25\ mm,\ D_{h,valve} = 2.4\ cm,\ AVA = 0.8\ cm^2$
Case 2: $D_{sinus} = 30\ mm,\ D_{h,valve} = 2.4\ cm,\ AVA = 0.8\ cm^2$
Case 3: $D_{sinus} = 35\ mm,\ D_{h,valve} = 2.4\ cm,\ AVA = 0.8\ cm^2$

**Fig 5. Mean aortic transvalvular gradient (mmHg) over a range of mean flow rate (ml/s) for an aortic valve area (AVA) of 0.8cm², a constant hydraulic valve orifice area (D$_h$) of 2.4cm and a κ–value (energy loss coefficient) as determined by a sinus diameter (D$_{sinus}$) of 25mm, 30mm and 35mm respectively.**

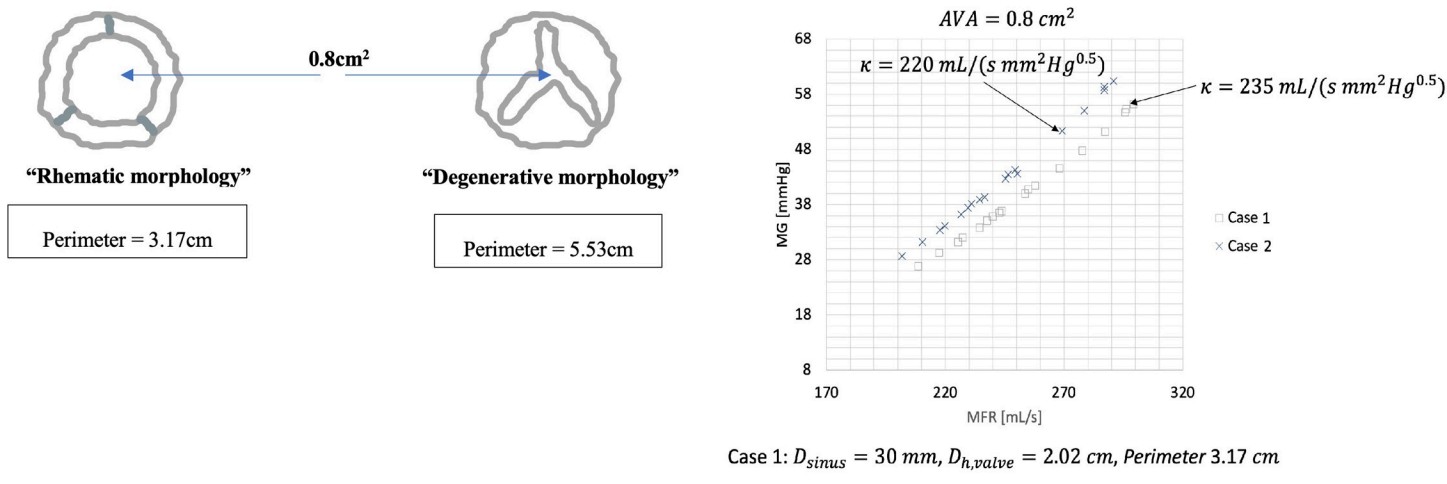

Case 1: $D_{sinus} = 30\ mm,\ D_{h,valve} = 2.02\ cm,\ Perimeter\ 3.17\ cm$

Case 2: $D_{sinus} = 30\ mm,\ D_{h,valve} = 1.2\ cm,\ Perimeter\ 5.53\ cm$

**Fig 6.  a:** A typical rheumatic aortic stenosis valve orifice morphology vs a degenerative aortic stenosis valve orifice morphology. Both valves have similar 'hemodynamic orifice areas', however, because of significant differences in perimeter, the 'hydraulic orifice' areas will be noticeably different. **b:** Mean transvalvular gradient–mean transvalvular flow rate relationship for a hemodynamic aortic valve area of 0.8cm2, a sinus diameter of 30mm but with a K (energy loss coefficient) as determined by a typical rheumatic aortic valve morphology vs a degenerative aortic valve morphology (MG–mean transvalvular gradient; MFR–mean transvalvular flow rate).

literature that examines the relationship between MG and SAP is limited in as far as it does not account for, or document, the full extent of factors that influence the magnitude of the effect. This likely results in findings that would only be applicable for very specific physiological boundary conditions that happened to prevail at the time of the experiment.

Current guidelines recommend that AS severity should ideally be quantified in a normotensive state. Because of the known impact of hypertension on MG, this appears to be a fair general recommendation. The findings of the current work suggest that whereas this may be very relevant in some patients, it may have no impact on the MG in others.

Changes in BP predominantly mediates its effect on MG via changes in MFR. The magnitude of these changes is intimately dependent on the baseline MFR as determined by the LV contractile state, end diastolic volume and the ejection time, as well as the severity of the AS. It is therefore prudent to consider each of the determinants of the MFR state of the LV when predicting the response that changes in BP may have on MFR, and subsequently on MG.

Other than MFR, the other important determinant of transvalvular gradients is the combined energy loss coefficient ($\kappa$), which represents the sum of contraction losses, friction losses at the valve orifice, and sudden expansion losses as blood exits the valve. In the simplified Bernoulli equation that is used worldwide on a daily basis, $\kappa$ has conveniently, but also somewhat arbitrarily been ascribed a number of 4. This represent a gross oversimplification, and the limitation of such an approach is unmasked in conditions were turbulent flow and stenoses are added to the system, such as we find in AS. Sudden expansion losses are the most important component of $\kappa$, and is influenced by the ratio of the true hydraulic valve area to the sinus dimension. In the current work we have demonstrated the impact of changes in sinus dimension and valve morphology on the expected MG—MFR relationship.

The current work appeals to the clinical community to perform further in–vitro and in–vivo studies, investigating the relationship between hypertension and MG. It would be prudent in such research to consider the MFR, the LV elastance, the AVA, the left ventricular diastolic volume, ejection times and a patient specific $\kappa$ that prevails at the time of testing to ensure generalizability of experiment findings.

## Conclusion

The impact of hypertension on the mean transvalvular gradients is complex. Using in–silico modelling, we identified and quantified the factors that influence this relationship. This work lays the groundwork for much needed and comprehensive in–vivo and clinical studies.

## Limitations

The experimental data presented in this research displays the utility of using sophisticated mathematical modelling of flow dynamics across a spectrum of predefined and selected conditions. This allows one to describe response behaviors that are not limited to the known and unknown characteristics that may be present in a specific subgroup of human or animal subjects. However, despite the comprehensiveness of the modelling, the current findings were not verified in–vivo. Encompassing the complexity of human physiological responses may remain elusive to mathematical models, and hence, such findings should be regarded as hypothesis generating and should pave the way for subsequent in–vivo validation studies. It should be noted, that despite these known limitations, the mathematical basis for the modelling is well validated from previous publications. A limitation in the design of the current modeling includes lack of consideration for percussion waves. The impact of percussion waves on MG across the spectrum of valvular stenoses and LV elastances is yet to be fully elucidated. The input variables for the modeling were derived from prior publications and uses an average

sized middle-aged male as reference. The extrapolation to different genders, body surface areas and ages is thought to be a reasonable assumption but will need future validation. Finally, the work should be regarded as hypothesis generating using idealized conditions, that requires external validation in in-vivo studies.

## Supporting information

**S1 File. Mathematical model development.**
(DOCX)

## Author Contributions

**Conceptualization:** Jacques Liebenberg, Ryno Laubscher, Philip Herbst.

**Data curation:** Jacques Liebenberg, Ryno Laubscher.

**Formal analysis:** Philip Herbst.

**Funding acquisition:** Anton Doubell.

**Investigation:** Jacques Liebenberg.

**Methodology:** Jacques Liebenberg, Ryno Laubscher.

**Supervision:** Anton Doubell, Philip Herbst.

**Writing – original draft:** Jacques Liebenberg, Anton Doubell.

**Writing – review & editing:** Jacques Liebenberg, Philip Herbst.

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
