## [Decision Letter · Decision Letter 0]

30 Mar 2023

PONE-D-22-35684In-silico modelling of the impact of hypertension on mean transvalvular gradient in aortic stenosisPLOS ONE

Dear Dr. Liebenberg,

Thank you for submitting your manuscript to PLOS ONE. After careful consideration, we feel that it has merit but does not fully meet PLOS ONE’s publication criteria as it currently stands. Therefore, we invite you to submit a revised version of the manuscript that addresses the points raised during the review process. All issues raised are required. The authors should pay attention to english grammar and style.

We look forward to receiving your revised manuscript.

Kind regards,

Vincenzo Lionetti, M.D., PhD

Academic Editor

PLOS ONE

Journal Requirements:

2. Please amend either the title on the online submission form (via Edit Submission) or the title in the manuscript so that they are identical.

Reviewers' comments:

Reviewer's Responses to Questions

**Comments to the Author**

1. Is the manuscript technically sound, and do the data support the conclusions?

Reviewer #1: Partly

2. Has the statistical analysis been performed appropriately and rigorously? 

Reviewer #1: N/A

3. Have the authors made all data underlying the findings in their manuscript fully available?

Reviewer #1: Yes

4. Is the manuscript presented in an intelligible fashion and written in standard English?

Reviewer #1: Yes

5. Review Comments to the Author

Reviewer #1: The authors provided insights to elucidate the interaction between hypertension and mean gradients in AS.

1. Major concern. Figures.

a) Is Figure 1 missing?

b) Figure 3. Labels to axes (both x and y - MFR and SAP) should be added to plots (with units of measurement). Grid on the plot with AVA=1.5 cm2 should be in harmony with other plots.

c) Figure 4. Labels to axes (both x and y - MG and MFR) should be added to plots (with units of measurement). Grid on the plot with AVA=0.8 cm2 should be in harmony with other plots. Elastance slopes should be added numerically to the graphs in support of the result: ".....LV elastance has no direct bearing on the MG…". What is the “central illustration” mentioned in row 231?

d) Figure 5. Units of measurement should be added. Title (AVA = 0.8 cm2) should be added as the previous figures. Legend for different energy loss coefficients (green, orange and purple) should be added. Figure 5 differs from figure 4 due to the energy loss coefficient. Although the trend is the same, the values change importantly (Figure 4 - minimum MG > 30 and maximum MG > 65 VS Figure 5 - minimum MG < 25 and maximum MG < 48). With which aortic sinus dimension do we approach the values in Figure 4? Maybe this case should be added to figure 5.

e) Figure 6b. Units of measurement. Title (AVA = 0.8 cm2). Has a new LV elastance been added? What is the value of E? Why was this elastance not added to the previous figures?

2. Interesting the strong changes in MG and MFR values with AVA less than 1.5 cm2. It would be interesting to see the results with other AVA values (i.e. 1.4 - 1.2 - 1.1 - 0.9 - 0.7 - 0.6).

3. The reviewer recommends softening some sentences out of caution. i.e.

Row 267 – “We have demonstrated that the impact of hypertension on MG is mediated through changes in MFR”

Row 226 – “it is demonstrated that the LV elastance has no direct bearing on the MG”

Row 293,306

6. PLOS authors have the option to publish the peer review history of their article (what does this mean?). If published, this will include your full peer review and any attached files.

Reviewer #1: No

---

## [Author Response · Author response to Decision Letter 0]

24 Apr 2023

To the Editor 

Thank you for the opportunity to respond to the excellent comments of the reviewer. We are pleased with the insight and attention to detail that went into the review. The comments are well received, and the revisions add value to the overall paper. 

Please find attached our response to the reviewer’s comments. We have also submitted a ‘markup retained’ copy of the manuscript as well as a ‘markup removed’ version via the submission portal as requested. 

Response to reviewers comments: 

1a. Is Figure 1 missing? 

It seems that our labelling of the figures led to some confusion. In the submitted version of the manuscript, Figure 1 is the central illustration, and can be seen on the last page of the auto-generated PDF submission document. 

To clarify this in the revised manuscript, we have opted to label the central illustration as “Central Illustration”, with subsequent figures numbered sequentially in the order they appear in the paper.

1b. Reference to Figure 3 (now figure 2)

Comments are well received, and the suggested changes have been made. 

1c. Reference to Figure 4 (now figure 3) 

Thank you for the remarks. The changes have been made as suggested. The reviewer refers to the central illustration in row 231 – please see comment at 1a. 

1d. Reference to figure 5 (now figure 4)

An updated version of this figure was uploaded. In the original figure, the peripheral compliance was altered to change the mean flow rate range as part of follow up modeling on this topic. This was incorrectly uploaded with the changed flow rate. The new figure is now in harmony with the flow rate ranges seen in figure 3. Since the flow rate ranges are now in harmony, we have opted not to add the case in figure 3 to figure 4 to avoid confusion. 

1e: Reference to figure 6b (now 5b) 

An updated version of this figure was uploaded. Similar to in the comments in 1d, the uploaded figure was part of a subsequent modeling where peripheral compliance was altered to assess the effect of higher flow rates and was incorrectly loaded with the higher flow rate. The flow rate range is now in harmony with the other figures and the elastances used in figure 3 (up to a maximum of 2.5). 

2. Reference to changes in MG and MFR with AVA’s less than 1.5cm2

Indeed, an exponential increase can be expected with progressively smaller valve areas. This is because of the quadratic relationship between the aortic valve area and the subsequent flow. The illustrated valve areas were chosen as they represent important clinical parameters defining a normal valve, a moderately, severely, and very severely stenosed valve. 

3. Suggestion to softening certain statements: 

 Row 267 – “We have demonstrated that the impact of hypertension on MG is mediated through changes in MFR.

The authors are of the understanding that, at least the predominant mechanism for hypertension induced changes in mean gradient is related to the impact of hypertension on the mean flow rate. This point was argued using first principles (Reference to Line 89-95). 

Alternative pathomechanistic pathways through which hypertension influences MG is not known to us and not illustrated in the current work. However, it remains unclear whether acute changes in blood pressure affects aortic sinus diameter acutely, and given this uncertainty, the sentence was modified as suggested. 

 Row 226 – “It is demonstrated that LV elastance has no direct bearing on the MG”

This sentence, as stated, could lead to misconceptions, and was modified to reflect the fact that LV elastance appears to have no independent impact on MG, other than through impacting the MFR, and hence the MG. In other words, if a compensatory mechanism were to keep the MFR constant, we would not expect a change in elastance to have a bearing on the MG. 

 Row 293 - Changes in BP mediates its effect on MG via changes in MFR

See comment at row 267. This sentence was similarly amended. 

 Row 306 - Sudden expansion losses are the most important component of κ , and is influenced by the ratio of the true hydraulic valve area to the sinus dimension. In the current work we have demonstrated the impact of changes in sinus dimension and valve morphology on the expected MG - MFR relationship (Row 304 – 307) 

We are not sure what the reviewer suggested to change in the statement. The statement in the first line was supported by a previous publication (referenced in the manuscript). The subsequent sentence refers to study findings that are supported by the current data. 

We hope that this brings clarification to the questions raised and again thank the reviewer for his insights, which have improved the document.

Your sincerely 

J. Liebenberg

---

## [Decision Letter · Decision Letter 1]

22 May 2023

In-silico modelling of the impact of hypertension on the mean transvalvular gradients in aortic stenosis

PONE-D-22-35684R1

Dear Dr. Liebenberg,

We’re pleased to inform you that your manuscript has been judged scientifically suitable for publication and will be formally accepted for publication once it meets all outstanding technical requirements.

Kind regards,

Vincenzo Lionetti, M.D., PhD

Academic Editor

PLOS ONE

Additional Editor Comments (optional):

Reviewers' comments:

Reviewer's Responses to Questions

**Comments to the Author**

1. If the authors have adequately addressed your comments raised in a previous round of review and you feel that this manuscript is now acceptable for publication, you may indicate that here to bypass the “Comments to the Author” section, enter your conflict of interest statement in the “Confidential to Editor” section, and submit your "Accept" recommendation.

Reviewer #1: All comments have been addressed

2. Is the manuscript technically sound, and do the data support the conclusions?

Reviewer #1: (No Response)

3. Has the statistical analysis been performed appropriately and rigorously? 

Reviewer #1: (No Response)

4. Have the authors made all data underlying the findings in their manuscript fully available?

Reviewer #1: (No Response)

5. Is the manuscript presented in an intelligible fashion and written in standard English?

Reviewer #1: (No Response)

6. Review Comments to the Author

Reviewer #1: The current version of the article is acceptable and has sufficiently addressed the revisions. I have noticed a few improvements that could be made to the "Central Illustration".

In both graphs on the right, it would be beneficial to modify the letter ‘x’ from "AVA = x cm2", as the variable "x" is already used in the plot and might cause confusion (i.e. with x [ml/ (s mm2 mmHg]).

In the "Therapy-Induced Changes in BloodPressure" panel, add the space between ‘blood’ and ‘pressure’.

Overall, the current version of the article is satisfactory.

7. PLOS authors have the option to publish the peer review history of their article (what does this mean?). If published, this will include your full peer review and any attached files.

Reviewer #1: No

---

## [Editor Report · Acceptance letter]

30 May 2023

PONE-D-22-35684R1 

In-silico modelling of the impact of hypertension on the mean transvalvular gradients in aortic stenosis 

Dear Dr. Liebenberg:

I'm pleased to inform you that your manuscript has been deemed suitable for publication in PLOS ONE. Congratulations! Your manuscript is now with our production department. 

Kind regards, 

on behalf of

Prof. Vincenzo Lionetti 

Academic Editor

PLOS ONE